# *In Vitro* Methods to Study Colon Release: State of the Art and An Outlook on New Strategies for Better *In-Vitro* Biorelevant Release Media

**DOI:** 10.3390/pharmaceutics11020095

**Published:** 2019-02-22

**Authors:** Marie Wahlgren, Magdalena Axenstrand, Åsa Håkansson, Ali Marefati, Betty Lomstein Pedersen

**Affiliations:** 1Department of Food technology engineering and nutrition, Lund University, P.O. Box 124, 221 00 Lund, Sweden; magdalenaaxenstrand@gmail.com (M.A.); asa.hakansson@food.lth.se (A.H.); ali.marefati@food.lth.se (A.M.); 2Ferring International PharmaScience Center (IPC), Kay Fiskers Plads 11, 2300 Copenhagen, Denmark; Betty.Pedersen@ferring.com

**Keywords:** *in vitro* systems, colon delivery, colon microbiota

## Abstract

The primary focus of this review is a discussion regarding in vitro media for colon release, but we also give a brief overview of colon delivery and the colon microbiota as a baseline for this discussion. The large intestine is colonized by a vast number of bacteria, approximately 10^12^ per gram of intestinal content. The microbial community in the colon is complex and there is still much that is unknown about its composition and the activity of the microbiome. However, it is evident that this complex microbiota will affect the release from oral formulations targeting the colon. This includes the release of active drug substances, food supplements, and live microorganisms, such as probiotic bacteria and bacteria used for microbiota transplantations. Currently, there are no standardized colon release media, but researchers employ in vitro models representing the colon ranging from reasonable simple systems with adjusted pH with or without key enzymes to the use of fecal samples. In this review, we present the pros and cons for different existing in vitro models. Furthermore, we summarize the current knowledge of the colonic microbiota composition which is of importance to the fermentation capacity of carbohydrates and suggest a strategy to choose bacteria for a new more standardized in vitro dissolution medium for the colon.

## 1. Introduction

Most drugs are adsorbed in the upper GI-tract. However, for a range of therapeutics, food supplements and probiotics delivery to the colon is important. This is especially true for a range of inflammatory diseases of the GI-tract such as Crohn’s disease and Ulcerative colitis [1,2,3,4,5,6] but also for other diseases in the colon that would benefit from a local treatment including colon cancer [7,8], enteric nematodes [9] and enzyme replacement therapies [10]. Other areas of interest are colon delivery of proteins and peptides [11,12], probiotic bacteria [13,14] and microbiota replacement therapies [15]. In these cases, formulations are employed aiming at colon-targeted delivery and at the same time avoiding release in the upper GI-tract. Thus, these formulations can protect the active ingredient from degradation in the stomach and small intestine. 

There are a range of colon delivery systems, but it is not within the scope of this article to do an in-depth review of these. Instead, we recommend some of the literature referred to in the next sections and the following general reviews [2,16]. The principle of these formulations will however define the demands on colon delivery in in vitro methods used as they are dependent on different triggers of release. The main release mechanisms described in the literature are based on a few different triggers.

*Degradation by the colonic microbiota:* One strategy is to use excipients that are degraded by the microorganisms in the colon. Two of the most common classes of excipients used are carbohydrates and azo compounds. The bacteria of colon produce a large repertoire of enzymes, among which some are able to digest complex carbohydrates that have escaped digestion in the small intestine. These include enzymes such as amylases, pectinases and β-d-galactosidases to mention a few. Such enzymes have the ability to hydrolyze polysaccharide-specific bonds. Polysaccharides that can be used as coatings for this purpose are resistant starches, guar gum, pectin, dextran, inulin, and chitosan [10,13,17,18,19,20,21]. These are all molecules that can only be degraded by the microorganisms in the colon. Another colon bacterial-induced release is azore-reductive bacterial activities [22]. For a more thorough overview of these systems, we recommend the review by Sinha and Kumria [23]. One drawback with this principle is that some patients especially with inflammatory disease have a different colonic microbiota that could affect these extracellular enzymes [2,24].

*Time-controlled release:* These are primarily based on slow eroding or dissolving polymer films or matrixes [25,26,27]. In such systems, the polymer or wax (primarily natural waxes have been used) responsible for the release should not be sensitive to pH, enzymes or other components of the lumen such as bile salts. These types of formulations will be partly affected by transport of water and components in the GI tract fluid into the formulation and thus, the kinetics of this transport are expected to be highly affected by the increased viscosity of the lumen [28] when water is adsorbed in the colon, as has also been seen for some formulations [29]. The effect flow-behavior in the intestinal lumen has also been seen to influence the uptake of nutrients from food and this has been extensively discussed in a review by Takahashi [30]. Furthermore, the rate of gastric emptying will also influence if the time delayed formulations will release in the colon. It is well known that the gastric emptying time will vary considerable for larger objects such as a tablet. 

*pH controlled release:* These formulations are based on the difference in pH along the GI-tract and are based on film coating that dissolves at different pH [31,32]. In order to secure release in the colon, some employ two or more films with different properties [32]. In some cases, pH dependence and time dependence are combined [33,34,35,36]. The main drawback of these systems is the large intra-individual variation in pH along the GI-tract, especially for patients with GI-tract diseases. A recent expert opinion concluded that due to the variation of pH in the GI-tract, colon delivery formulations that is only dependent on pH difference has a major risk of both premature drug release and no release at all [37]. 

*Pressure-controlled delivery:* This is based on the fact that the high viscosity of the lumen combined with the smooth muscle contractions of the colon assert a mechanical pressure on the formulation that ruptures a film coating or a capsule [38,39]. Primarily for capsules, the film strength of the capsule can be designed in such way that it ruptures at these pressures [16]. The drawback is that the main colonic motor pattern, pan-colonic pressurizations, can be severally reduced for example in patients with chronic constipation [40]. Furthermore, the less frequent high-amplitude propagating sequences that might be especially important for these formulations are only present after meals and not in all individuals studied [41].

*Nanoparticle drug delivery systems:* Various types of advanced nanoparticle formulations for colon delivery have been investigated by numerous researchers. These can for example be based on poly(lactic-*co*-glycolic acid) (PLGA), lipids, chitosan or silica, and these nanoparticles could be targeting inflamed mucosa. More details can be found in the reviews by Zhang et al. [42], Hua et al. [43] and by Vass et al. [44]. These types of formulation have the potential to deliver better treatment to the patients but there is a need for more attention to the practical design of the final dosage forms that successfully deliver the nanoparticles to the colon.

All of these formulation principles have their pros and cons and few of them have been found to deliver drugs to the whole colon. Thus, there is still a need to develop new colon-targeting formulations. In both early and late formulation development phases, there is a need for suitable and biorelevant in vitro media for evaluation of the various formulation concepts. Furthermore, during the formulation development of chosen concepts there is a need to evaluate the release profiles from different formulation designs and manufacturing principles.

The recommended dissolution media in the European and US Pharmacopeias are focused on simulating the upper GI tract. There is no Pharmacopeia in vitro dissolution method for colon delivery as such and the release media employed by several researchers for the evaluation of colon release differs a lot. It is the purpose of this review to discuss the benefit and draw backs of different colon release-testing media and also to suggest novel release media based on microorganisms present in the colon. In discussing in vitro methods, we also have to look at the whole passage through the GI-tract. Thus, this review will briefly describe the GI-tract and some of the dissolution media used to resemble the upper parts of the GI-tract.

Also within food research there is an interest in understanding release and digestion, especially concerning functional foods which are developed with the aim of promoting health and preventing diseases such as obesity, cancer, diabetes, neurodegenerative diseases (Alzheimer and Parkinson), and others [45]. These systems are developed in a way that the bioactive ingredient is protected while controlling and targeting the release to the specific location in the human gastrointestinal tract where they act [45]. The bioactive ingredients include polyphenols, carotenoids, fatty acids, proteins, peptides, amino acids, vitamins, minerals, and even live probiotic bacteria [46]. The challenge of inclusion of these bioactive ingredients is that they are susceptible to the conditions dictated by processing, storage and digestion that may lead to a decline in bioaccessibility as well as bioavailability. As a result, and in order to see the fate of bioactive compounds in human gastrointestinal conditions, simulated in vitro and in vivo digestion models are frequently used. In this review, we will do a comparison between the methods used in pharmaceutical development with those used for food research.

Even though the dissolution media are the focus of this review, it is important to bear in mind that an in vitro release profile from an oral dosage form is determined by the choice of dissolution method which comprises the (1) dissolution medium (composition and volume), (2) physical apparatus design, and (3) apparatus settings, that determine the hydrodynamics during release from the dosage form.

The review will discuss the pros and cons of most existing dissolution medias used for colon release. However, the main focus will be on media containing microorganisms. These medias are complex but should be the most biorelevant ones as the microbiota is the component that primarily governs the environmental conditions in the colon. Using live microorganisms ensures that the media contains a range of enzymes, and that these enzymes are constantly renewed by the bacteria. This is especially important when the release mechanism is based on degradation by the colonic microbiota. One could also suspect that a complex live microbiota to some extent could adapt to the nutrients given for example if carbohydrate-based formulations are investigated. One of the advantages of using a complex media is that different formulations, for example formulations based on different carbohydrates, can be investigated. 

## 2. The GI Tract

### 2.1. The Upper GI-Tract

Drugs that are administrated orally are exposed to a changing environment, and if a drug is intended to target the colon, it will be exposed both to the upper part of the GI tract and colon. Figure 1 illustrates the changing conditions encountered during passage through the GI-tract. One of the key uses is the changing pH in the GI tract. In the stomach the pH normally varies between 1 and 3.5. Higher gastric pH up to 4.6 has been observed in healthy human subjects by Koziolek et al. [47] and higher gastric pH can be expected in patients receiving gastric acid blocker therapy and in the elderly it can be elevated up to pH > 5 [48]. Traditionally, oral formulations that are designed to avoid gastric release are coated with a polymer that is insoluble at low pH but soluble at higher pH, the so-called enteric coatings. 

The content of the stomach is released into the small intestine. The surface area of the small intestinal mucosa is large due to the presence of villi and microvilli’s that increase the contact area between the lumen and the epithelial cell wall which promotes drug absorption [49]. The pH of the small intestine ranges between 5.5 and 7.5 [50,51]. The lumen of the intestine also contains numerous enzymes as well as bile salts. The enzymes include hydrolases and proteases such as trypsin, chymotrypsin and carboxypeptidases as well as lipases and amylases. Bile salt are reabsorbed in ileum and about 95% of the bile salts are recirculated [52]. The concentration of bile and enzymes show considerable individual variation and vary between the fasted or fed state. Compared to the colon, the upper GI-tract contains only a minor amount of bacteria, of around 10^3^–10^4^ CFU/mL [53,54], and thus, enzyme from bacteria play a minor role compared to the endogenous enzymes in the small intestine. 

### 2.2. The Physiology of Colon

The main function of the colon is uptake of water and different ions from the colonic content, and it serves as a storage and compaction space for feces The microorganisms that reside in the colon have important consequences for human health, being active fermenters of undigested polysaccharides, and the community composition and activities are known to be strongly influenced by the dietary carbohydrate content [59,60,61]. The addition of probiotic bacteria has for example been seen to affect such diverse disorders as stress [62], colonic carcinogenesis and hepatic injuries [63] and obesity management [64,65]. Another emerging topic is fecal transplantation focusing on not one microorganism but the whole microbiota from healthy donors [66]. 

The colon makes up the final 1.5 m of the GI tract and can be further divided into smaller parts such as the caecum, ascending, transverse and descending colon, see Figure 1. The pH of colon varies between 5.7 and 6.7 along the large intestine. It is lowest in the ascending colon and then it increases. It should also be noted that for patients with diseases such as Crohn’s disease and Ulcerative colitis, the pH can often be reduced compared to the pH in healthy subjects [2]. 

Colon is a strictly anaerobic environment and its microbiota is a dense and complex community comprised by mainly obligate anaerobe bacteria. Compared to the stomach and small intestine, the colon harbors a much larger population of bacteria with up to 10^11^ bacteria per gram of intestinal content. The microbiota of the colon is different in different regions. Close to the epithelial barrier, there is a nearly bacterial free zone of mucus and then the microorganisms are spatially organized with different compositions of bacteria closer to the mucin than those in contact with lumen. The microbiota of the mucosa also varies along the colon [67,68]. These microorganisms produce a large panel of enzymes that are active in the breakdown of dietary fibers that escape digestion in the small intestine. Tasse et al. [69] used coupled functional screens and sequence-based metagenomics to identify highly prevalent genes encoding enzymes that are involved in the catabolism of dietary fibers by the human gut microbiome and found that they produce a range of carbohydrate digestive enzymes such as beta-glucanase, hemicellulase, galactanase, amylase, or pectinase [69]. The bacteria of the colon also metabolize the bile salts that has not been taken up in ileum. Thus, the bile salt composition is different in the colon compared to the small intestine, see Table 1. The pancreatic enzymes are also digested in the colon and amylase was for example found to decrease by 50% while protease decreased by 70% [70].

Compared to the upper GI-tract, most of the nutrients in lumen have already been adsorbed when reaching the colon [71]. However, the amount of free fatty acids is increased from around 6–8 mM in ileum to 32–29 mM in cecum. In a fed state, the nutrient content of cecum was around 6 mg/mL of protein and 10 mg/mL of soluble carbohydrates [71]. This can be compared to the content in jejunum where the protein content of 1 mg/mL in the fasted state and 5 mg/mL in the fed state has been determined using the Loc-in-Gut technique [72]. 

## 3. In Vitro Release Methods

A dissolution test is a key tool for determining the release profile during the development of an oral drug product and it is especially important for controlled release formulations. Although research on dissolution processes goes back to the 1890s, when the topic was mainly studied by physical chemists, it took many years before its applicability and importance in pharmaceuticals was implemented. Until the 1950s, it was believed that the bioavailability of drugs was only dependent on disintegration of the tablet [73]. However, in 1957, Nelson related theophylline blood concentration to dissolution rate [74]. His work was followed by several other studies investigating the correlation between dissolution and bioavailability. During this time, it was discovered that the formulation of the drug could affect the pharmacological effect. In the 1970s, large differences were discovered between different formulations and brands of the drug digoxin [75],which further supported the theory that the release from formulation influences bioequivalence. Cases of drug toxicity and reduced drug effects as a result of alternation of an excipient were observed [76]. Incidents like these led to the realization of the importance of dissolution studies for the purpose of quality control and the introduction of dissolution requirements in pharmacopeias of dissolution data for tablets and capsules. Additionally, standard dissolution methods were introduced in the Unites States Pharmacopeia, National Formulary [73] and also in other Pharmacopeias such as those for Europe and Japan. The physical designs of the most frequently employed standard pharmaceutical apparatus are described in the Pharmacopoeias. These are the Basket (USP 1) and Paddle (USP 2) apparatus, the reciprocating cylinder apparatus (USP 3) and flow through apparatus (USP 4). The Pharmacopoeias also recommend the range of settings that determine the hydrodynamics during the dissolution test. More biorelevant apparatus have been suggested by several researchers and some of these were reviewed together with the standard dissolution equipment by Kostewicz et al. [77]. Another relevant apparatus is the Dynamic Colon Model that is designed to mimic the architecture, physical pressures and motility patterns in the proximal colon [78].

One key use of in vitro release testing is to try to use it for prediction of *in vivo* uptake, the so called in vitro–in vivo correlation (IVIVC). There are three different levels of IVIVC [79,80]. In Level A correlations, all data collected (both dissolution and blood plasma concentration) is used to create a point-to-point relationship. It is considered to be the highest level of correlation and is usually, but not always, a linear relationship. This direct relationship between in vivo and in vitro data can be used to predict in vivo performance. In Level B, all the obtained data is used just as in Level A correlations, however, a point-to-point relationship is not created, instead the mean in vitro dissolution time is related to the mean in vivo dissolution time or mean in vivo residence time. A Level C correlation is a single point correlation where only one point from the in vitro dissolution profile is related to an in vivo parameter such as maximum blood plasma concentration, Cmax. No prediction of the in vivo blood concentration profile can be made using this type of correlation. Developing a meaningful IVIVC is, however, always challenging since there are many factors to consider, such as transit time and composition of the gastric and intestinal content including pH, bile salts and enzymes that affects the release and uptake of a drug [81]. 

When conducting in vitro studies, the choice of dissolution media is important. There are several strategies on how to choose the media going from very simple solutions, in some cases even just water, to very complex ones that are more biologically relevant [80]. It is important to point out that the purpose of a dissolution method is not to simulate the exact conditions in vivo, which of course is impossible. It is, however, important that the dissolution method resembles the characteristics of the in vivo situation that determines the release profile. This does not mean that the method should not be as realistic as possible, but the goal is to be able to compare a system over time and between different laboratories. Without a reproducible way to do this, the test loses its meaning. Thus, there is often a tradeoff between highly realistic and complex media and the reproducibility. In designing a dissolution method, the purpose of the method often sets the degree of complexity. For dissolution methods that are going to be used in quality control, reproducibility and capability is in focus since it is more important to detect production variability than the biological relevance of the method. For research and development purpose as well as for establishing an IVIVC, the biorelevance of the dissolution media is of high importance. 

In the case of oral formulations, it is the GI tract that has to be simulated during the dissolution studies. Due to the variation in pH along the GI tract, the pH may be adjusted during the in vitro experiment to simulate the transit through the GI tract. This is important for understanding the interaction of the formulation with pH, the dissolution of acids and bases and to understand the survival of probiotics [14]. In the case of probiotics, the aim is to modulate the microbiota to give health benefits to the consumer. In most cases, probiotics are given as a cost supplement but with growing evidence of the effects on several diseases, the interest to use live bacteria also as a pharmaceutical product is increasing. The demands on such a pharmaceutical product will be different from most current probiotics and for oral products survival through the stomach and upper GI tract will be crucial for the probiotics to work properly. As pointed out by Quigley, there is a need for both better formulations and quality controls when it comes to many of these products [82]. For further reading on probiotics, we recommend some of the following recent reviews on the topic [82,83,84,85,86].

Both the stomach and small intestine can be simulated using the pharmacopoeia dissolution media. These media range from simple solutions with low complexity to solutions containing surfactants or enzymes which complexity wise can be categorized as medium to high, respectively (see Table 2). For the sake of this review, the media in Table 2 have been categorized with regards to a representation of either the fasted or fed state. Furthermore, it is worth noting that the USP <1092> [79] is suggesting the option that dissolution media may simulate the gastric and intestinal fluids more closely with regards to both the ionic strength and molarity of the buffers employed. 

For conducting dissolution tests of controlled release formulations, these are normally conducted in series. The European Pharmacopoeia recommends pH steps and timings for dissolution tests involving steps with increasing pH [87]. These steps start in acidic milieu (pH 1.0–1.5) representing the stomach and may end at the highest pH levels (pH 7.2–7.5) representing the lower parts of the small intestines. Typically, the formulation is left in gastric conditions for the approximated transit time through the stomach, usually 1 or 2 h [87,88]. This is followed by an adjustment of the solution to the conditions for simulated small intestinal juice. Normally, it is only the upper parts of the GI-tract that are studied during the release tests but for colon delivery also the release in simulated colon conditions has to be studied. These testing conditions do not include a step simulating the colonic fluid.

Important changes induced by food ingestion include an increase in pH, and stimulation of secretion of bile salts and pancreatic enzymes. For this reason, simulations of fasted and fed conditions have been developed by several pharmaceutical researcher and in-depth reviews covering both simulated gastric and intestinal dissolution media have been made by Reppas et al. [93] and Bergström et al. [50]. For example, Vertzoni et al. suggested a fasted state simulated gastric fluid (FaSSGF) which besides hydrochloric acid contains pepsin as well as low amounts of bile salt and lecithin [94]. Simulation of the fed state in the stomach has been achieved by using milk for example. In a study from 2013, Christophersen and colleagues used milk together with bile salt and lipase to simulate a fed condition in the stomach and duodenum [95]. Worth noting is that the fed state is more complicated to simulate since the environment in the stomach changes over time as the food is digested and emptied into the small intestine [96]. For the simulation of the small intestinal fluids, the pioneering work by Prof. Dressman has led to standardized dissolution medias for fasted and fed state in the small intestines. These fasted states simulated intestinal fluids (FaSSIF) and fed state simulated intestinal fluids (FeSSIF) are extensively discussed in the review by Markopoulos et al. [97]. Biorelevant dissolution media are categorized by Markopoulos et al. in four levels. At level 0 media, only the difference in physiological pH along the GI tract is taken in to consideration while for level I, the distinction between fasted and fed is done based on both pH and buffer capacity of the media. At level II, the difference in bile components, lipids and their digestion products, and the osmolality are also considered, and in the most complex versions of level III, considerations such as changes in viscosity and digestive enzymes are included [97]. Another important aspect to consider when constructing biorelevant dissolution tests is that gastric emptying occurs less frequently when food is ingested which is why the gastric simulation step should be longer when simulating a fed state compared to a fasted state [50]. 

Interestingly, there are actually two different traditions in developing complex release media for simulating the GI tract, one based on pharmaceutical research and one based on food research. The perspective of these two traditions varies of course as the food research primarily focuses on the digestion of food components while pharmaceutical research focuses on the release of the active compound. For example, food research might include the chewing of the food, a part that is seldom relevant for pharmaceuticals. This means that also the enzymes from the mouth are included in such studies [98]. The in vitro models used in food research can be static [99] or dynamic [100,101] and batch or continuous [102]; however, regardless of the nature of the methods, all of them consist of simulated oral (mouth), gastric, and intestinal compartments and may or may not include the colon. Due to the complexity of these models, they are often adopted to the specific conditions of each system. In order to do so, the system is categorized to either protein-based, lipid-based or carbohydrate-based systems and the digestive conditions are chosen for the system being analyzed. In this way, if a protein-based delivery system is being evaluated, the main focus would be the gastric and intestinal steps where the corresponding proteases exist (i.e. pepsin in the stomach, trypsin and chymotrypsin in the small intestine). In a similar way, when lipid-based delivery systems are being evaluated, gastric and intestinal lipases are included; however, the focus is more on the intestinal lipolysis. Finally, if a carbohydrate-based delivery system is being analyzed, the oral and intestinal amylases are to be included. In some cases for simplification of the in vitro evaluation, whatever passes the oral, gastric and intestinal steps is considered to be delivered to the colon [103,104].

Many of the in vitro methods that are used for food research are more complex than the traditional pharmaceutical models. In the early 1990s, Molly et al. [105] developed a comprehensive dynamic model called “the simulator of the human intestinal microbial ecosystem” (SHIME). Through a multi-chamber system controlled by a computer, simulation of the conditions in the stomach, duodenum, jejunum, ileum, caecum, proximal, transverse, and distal colon, this model enabled scientist to control the concentrations of enzymes, bile, pH, temperature, feed composition, transit time, and anaerobic environment in each reactor.

Another model developed by Minekus, Marteau and Havenaar [101] was comprised of two separate models. TIM-1 simulates the conditions of the stomach, duodenum, jejunum, and ileum while TIM-2 simulates the large intestine controlled by a computer. An advantage of this model to SHIME was the possibility of controlled peristaltic movements and water absorption. It was later in that decade when Macfarlane, Macfarlane and Gibson [102] developed the three-stage continuous system which simulates the proximal, transverse and distal colon (differing slightly in pH and volume) containing fecal microbiota of healthy individuals and the system was then, in the late 1990s, validated against sudden death victims. The results of these types of evaluations have shown that the majority of carbohydrate breakdown and short-chain fatty acid production occurred in the proximal part of the colon while amino acid metabolism occurred in the transverse and distal colon [102]. Although these methods were developed for food and probiotic applications, they have since also been used in pharmaceutical research [106,107].

In addition to those models, the Dynamic Gastric Model (DGM) developed in the former Institute of Food Research in Norwich, UK as well as the Human Gastric Simulator (HGS) developed at UC-Davis, can be mentioned [108]. Similarly, simulated intestinal models were developed to investigate mass transfer from the luminal side [109]. Despite their potentials, these models have limitations due to limited accessibility worldwide and therefore, a consensus static digestion method was developed within the COST (European Cooperation in Science and Technology) Action, the so called InfoGest method [99,110]. The InfoGest method consists of a simulated upper GI (mouth to small intestines) method. Although this method is very good in providing details on the type and activity of the enzymes, ratios of digestive fluids, pH, ionic strength, and time, it is still very complicated, laborious and time-consuming.

Finally, one of the more advanced in vitro systems developed has been in the area of understanding how different components can affect the human gut microbiome. This in vitro microbiome modulation is performed in a three-stage culture system GIS2 simulator and the effect of the components on the microbiota is analyzed by determining the microbiological fingerprint using qPCR analysis [111]. 

### Colon Release

Most oral pharmaceutical formulation dissolution tests covering the stomach and small intestine are sufficient. However, for drugs that are intended to target the colon, an additional step simulating the large intestine needs to be added. This has proven to be challenging as the colon is a very complex environment and is therefore difficult to mimic. There is a severe lack of simple and relevant ways to perform dissolution tests for formulations that target the colon, and the methods available today have significant drawbacks.

Current methods available for simulating the large intestine include dissolution in buffers of relevant pH, the use of mixtures of enzymes that are known to be produced by bacteria in the colon, caecum content from animals (primarily rats), human fecal slurries, and fermentation of selected bacteria. Below, we give a short description of these methods, for a more thorough review on the first four methods, we recommend the review by Yang [112].

*Release in buffers:* This is the simplest and most reproducible system to use but its biological relevance is low. The system can be used for formulations based on pH triggered and time-dependent release. These buffers might be able to represent the key mechanism of release in vivo but the pH levels chosen are typically representing a mean in vivo pH and not the range of pH actually found in vivo. With regards to some formulations, the viscosity of the release media could also be of importance and as discussed in the introduction, the viscosity changes in the colonic lumen and could affect the transport of molecules and release from formulations. However, one has to be aware that many excipients are also degraded by the enzymes of the microbiota and to obtain relevant results, buffers can only be used when one is sure that no such degradation occurs. This being said, there are still several cases when pH control is the only method used [32,33,34]. Usually this is done in systems where no enzymatic break down would be expected but unfortunately also in cases where such break downs are an obvious risk.

*Use of enzymes present in the colon:* The use of an enzyme or mixtures of enzymes that are known to be produced by bacteria in the colon has the clear benefit of being considerably easy to handle, comparably inexpensive and has high reproducibility. The drawback is of course that even a mixture of enzymes is a very large simplification compared to the large range of bacterial enzymes in the colon. However, by choosing a high concentration of one enzyme, good correlation with fecal dissolution systems can sometimes be achieved as shown by Siew et al. [113]. In Table 3, we list some example of the enzymes that have been used in colon-simulating release media. The problem is still that different methods for colon delivery will be susceptible to different enzymes, and thus, the possibility to compare for example different carbohydrates for colon delivery is not straight forward using enzymatic release media. Instead, media such as feces and rat caecum, have been used.

*Use of rat caecum content*: To incubate the drug in a slurry of rat caecum content is one of the more common methods employed when the investigator prefer a more biologically relevant media. The benefits are that caecum can be treated in such a way that the anaerobic condition is maintained, and by adhering to a strict protocol of animal feeding, a rather high degree of reproducibility should be able to be obtained. Furthermore, the rats can be given a special diet to increase the bacterial species that are sensitive to the system investigated [123]. The drawback is primarily that the method involves the sacrifice of animals and results obtained from experiments on animals may not be directly applicable to humans due to microbiota compositional differences [124]. The microbiota in rodents might differ from humans, for example, lactobacilli is a major group in mice [125] but in humans lactobacilli are regularly present but in a smaller proportion [126].

Another interesting thing to note is that different research teams have been using quite different protocols for these experiments which means that it can be difficult to compare the results. Several groups have based their method on the work done by Prasad et al. [127], who used a USP bath with 4% of rat caecum in phosphate buffer pH 6.8, and to keep the system semi-anaerobic, the solution was bubbled through with CO_2_. There are slight variations to this method, for example, by using different pHs, 6.5 [128]–7.5 [129], using nitrogen in some steps of the method instead of CO_2_ [128], using variations in caecum content, 2%–10% [129,130], and varying the type of USP setup [128] or using sealed bottles [131]. In systems based on azo-structures, the intact or sonicated caecum (to release intracellular enzymes) have been mixed with co-factors such as benzyl viologen, NADP, glucose-6-phosphate dehydrogenase, and glucose-6-phosphate [122,132,133]. Caecum content from other animal species has also been used and for example guinea pig caecum was more similar to human fecal slurry in its breakage of azo-bonds than rat caecum [134]. 

*Using fecal slurries from animals or humans*: This has either been done using the same principle as for rat caecum, dispersing the sample in buffer and following the release in this solution during anaerobic conditions [135,136], or by using the fecal slurry to include a more complex fermentation step [4,100,137]. The main advantage of using fecal samples is that animals do not have to be sacrificed and it is possible to use human samples. One additional benefit is that it is possible to use fecal samples also from patients [138]. Using human feces makes it more relevant but the fact that it is fecal samples and not samples from the earlier parts of the colon is a problem. Primarily, these samples will not be harvested in anaerobic conditions, and thus, especially in the case of fermentation, there is a high risk that the anaerobic bacteria dominating in the colon are not viable in feces. The other problems are that the microbiota in the lower part of the colon is not the same as in the earlier parts of the colon and by using human samples there will be a large variation between samples from different individuals. Culturing the samples of feces or caecum for that matter before the addition of the formulation increases the activity of the system and can increase the rate of degradation as seen for example in the breakage of azo-bonds [134]. However, many of the studies are based on simply dispersing the feces (around 10%) in buffers with varying pH and incubating with the formulation as the only additional source of nutrient under anaerobic conditions [135,136,139,140]. Interesting to note, is that in most cases the pH employed is around 7.5. That is a pH closer to that of feces and not to the lower pH of the upper colon where most of the fermentation will take place in vivo. There are also, as discussed previously, several complex systems using coupled fermentation reactors to simulate the changing properties of the whole GI-tract or colon. This includes systems such as SHIME^®^, EnteroMix, Lacroix model and TIM-2, for a thorough review of these systems, please see Venema et al. [141]. When using a fermentation medium, the composition of the fermentation medium used for this purpose may be a simple but optimized and still complete medium, containing all principal components of a full-fermentation medium. However, it could also be of a more complex nature, mimicking a certain dietary habit, for example the prevailing Western-style diet [142,143].

*Using selected bacteria*: One of the earliest examples of using selected bacteria was the work by Karroute et al. [138], who used bifidobacteria, *Bacteroides* and *E. coli* to study release from pellets coated with Nutriose. In a study from 2015, Singh and colleagues used a probiotic mixture to simulate the large intestine when testing drug release from Sulfasalazine spheroids coated with different polysaccharides [144]. This system was compared to fecal slurries of humans, rats and goats, and the release profiles of carbohydrate-based formulations were rather similar, especially to the rat caecum content medium, even though the release in the probiotic release media was slightly slower [145]. The composition of the probiotic release media is presented in Table 4. In a similar study by Kotla et al. [146], a probiotic culture was once again used to mimic the colonic environment to examine drug release from 5-fluorouracil granules coated with polysaccharides. The species used are listed in Table 4. While no specific concentrations for each species were given by Kotal et al, the cultured media contained a total amount of 9.8 × 10^10^ CFU (colony forming units). In this study, the drug release profiles obtained from the tests with probiotic mixture, rat caecum content and human fecal slurries were to some extent comparable. In general, the probiotic mixture resulted in a slightly faster release than the other two media [146]. Unfortunately, the choice of bacteria was not thoroughly discussed in these articles although the authors suggested that these bacteria can simulate the conditions in the colon. The problem is, however, that although the colon contains some probiotic bacterial species as part of the microbiota composition, these are not the dominating species. In order to give suggestions for a more relevant media to simulate the colon, we present an overview of the literature that has investigated the microbiota of human colon. This has been done, especially with focus on carbohydrate degradation in the human large intestine. Based on this we suggest some key species that could be used for such systems. 

## 4. The Gut Microbiota

There are not many relevant studies on the human microbiota, i.e. human studies including identification and quantification of colonic bacteria on a species-specific level and based on a significant number of individuals. This suggests that although this particular microbial community has been studied extensively, it is still not very well understood. It appears that the general consensus is that the bacteria of the large intestine belong mainly to the phyla *Firmicutes* and *Bacteroidetes*. While several studies have indicated that species belonging to the *Clostridium* clusters IV and XIVa as well as to the *Bacteroidetes* phylum dominate the colonic flora, there is often a lack of examples of which bacteria dominate in the colon on a species-specific level. According to a study performed by Tap and colleagues where 16s rRNA sequencing was used to characterize the microbiota of colon, 79.4% and 16.9% of the sequences belonged to *Firmicutes* and *Bacteroidetes*, respectively [147]. This is further supported by the work of Yang et al., who by searching the Ribosomal Database Project identified bacteria that belonged to 13 phyla [148]; see Figure 2 for the most frequent phyla. However, it is not completely evident which species are dominant or most commonly occurring in humans. A reason for this is that there is a clear inter-individual diversity of the colonic microbiota among humans as a result of, for example, differences in dietary intake, age and presence of diseases, and the studies usually include only a few subjects [61]. It is therefore difficult to tell how representative the results are. As for differences in dietary intake, it has, for example, been observed that *Prevotella* spp. and other *Bacteroidetes* are more common in a group of rural African children while *Firmicutes* are more prevalent in Italian children. These variations were ascribed to differences in starch and fiber intake [60].

The above-mentioned study performed by Tap et al. [147], aimed at finding an intestinal microbiota phylogenetic core (a part of the microbiota that was common for more than 50% of the patients) among 17 healthy, both male and female, individuals. It was found that 2.1% of the detected operational taxonomic units (OTUs) were present in more than 50% of the individuals. Using quantitative PCR, they found that *Clostridium leptum* cluster IV, *Clostridium coccoides* cluster XIV and *Bacteroides*/*Prevotella* were the most dominating groups. The OTUs could be connected to species such as *Faecalibacterium prausnitzii* (present in 16 of 17 individuals), *Bacteroides vulgatus*, *Roseburia intestinalis*, *Ruminococcus bromii*, *Eubacterium rectale*, *Coprobacillus* sp. and *Bifidobacterium lognum*. It was found that the abundance of a certain OTU did not necessarily always relate to the frequency of observation. After applying a statistical model, it could be concluded that the 10 most frequent OTUs were related to *F. prausnitzii*, *Anaerostipes caccae*, *Clostridium spiroforme*, and *Bacteroides uniformis* among others [147]. Another study based on 16s rRNA stable isotope probing displayed a strong prevalence of sequences related to *Prevotella* spp. (*Bacteroidetes*), *Ruminococcus obeum* (*Clostridium* cluster XIVa), *R. bromii* (*Clostridium* cluster IV), *E. rectale* (*Clostridium* cluster XIVa), and *Bifidobacterium adolescentis* (*Actinobacteria*) [149]. Results also suggested that *R. bromii* might act as a primary starch degrader while the remaining species further degrade metabolites generated by *R. bromii.* This type of cross-feeding is not uncommon; on the contrary, it appears to be a central property of some anaerobic microbial communities [61].

In 1999, Suau and colleagues investigated the human bacterial colonic community by analyzing feces for adult males using cloned 16s rRNA sequences [126]. They found that 95% of the clones belonged to the *Bacteroides* group and to the species *Clostridium coccoides* and *Clostridium leptum*. The majority of recovered sequences did not belong to known cultivated microorganisms; however, sequences were found that correspond to bacteria such as *B. thetaiotaomicron*, *F. prausnitzii*, *B. uniformis*, *B. vulgatus*, *Eubacterium eligens*, *E. rectale*, and *R. bromii*. [126]. In yet another study, using 16S rRNA gene sequencing, Wang and colleagues detected and quantitated 12 bacterial species in human and animal fecal samples [150]. They found that *F. prausnitzii*, *Peptostreptococcus productus*, and *Clostridium clostridiiforme* had high PCR titers in all fecal samples. Additionally, high PCR titers were detected for *B. thetaiotaomicron*, *B. vulgatus*, and *Eubacterium limosum* in human adult fecal samples. The authors stated that their results are consistent with previous studies, however, some differences were detected when compared to cultured-based methods, and the findings were explained by some species being non-culturable, making enumeration using such methods dependent on how easily bacteria can be cultivated. In contrast, PCR can detect both unculturable and dead bacteria. Additionally, their PCR method detected the bacteria in situ while detection occurs after enrichment when using culture-based methods. Moore and Holdeman [151] used anaerobic tube culture techniques to evaluate the microbial composition in human feces from 20 healthy Japanese–Hawaiian males. They observed 113 different microorganisms using this method which accounted for 94% of all viable cells, and the microorganisms included *B. fragilis ss. vulgatus*, *F. prausnitzii*, *B. adolescentis*, *Eubacterium aerofaciens*, *P. productus*, *rectale*, and *R. bromii* among others. In another clinical trial, Wang et al. [152] found that sequences related to *Bacteroidetes* and *Clostridium* cluster XIV and IV dominated the clone libraries created from mucosal biopsies from the distal ileum, ascending colon and rectum. However, no examples of predominant species of the ascending colon were mentioned.

In the studies mentioned above, fecal samples were used for the characterization attempts. This is due to the fact that such samples are easy to collect, and it is believed that they represent the colonic microbiota well. However, it should be kept in mind that certain microorganisms may not be well- represented by fecal samples since they might be part of the colonic mucosal microflora. In a study by Hold and colleagues [153], 16s rRNA gene sequencing was performed on colonic tissue samples instead of fecal samples. The colonic tissue was taken from elderly people, DNA was extracted, and 16s rDNA was used for PCR amplification before it was cloned into vector plasmids. In this study, 46% of the clones that were analyzed belonged to the *Clostridium* cluster XIVa. Most sequences did not show the closest resemblance to current species-type strains, however, 16 of 51 sequences were loosely related to *Eubacterium ramulus*, *E. rectale*, *Roseburia cecicola* and *Eubacterium halii*; 14.5% of the sequences belonged to the *Clostridium* cluster IV where 6 out of 16 sequences belonged to *Ruminococcus* spp and 4 out of 16 showed the closest resemblance to *F. prausnitzii.* Sequences belonging to the genus *Bacteroides* constituted 26% of the analyzed clones with most of the sequences being closely related to *B. vulgatus* and *B. uniformis*. Additionally, sequence similarity to *Prevotella enoeca* was found in one of the three samples [153]. 

## 5. Can a Novel Media Based on the Microbiota be Developed?

A media based on chosen microorganisms instead of fecal samples would have several benefits. Primarily, the reproducibility of such a media would be considerably higher and if the bacteria are grown anaerobically the media would also to some extent be more relevant for the colon. However, the choice of bacteria, in regard to the variability of the microbiota, is not straight forward. The media should also have a relevant pH and the addition of other components such as human enzymes and bile salts could be considered. However, as the bile salts to a large extent are reabsorbed before the colon and the fact that the dominating enzymes are those produced by the bacteria, it could be possible to simplify the system by excluding these endogenous components. If a simulated colon dissolution media should be developed, there are several standards that should be fulfilled. Primarily, the bacteria chosen should be relevant for the colon but also relevant for the components in the formulation. Thus, bacteria that are known to produce extracellular enzymes that break down carbohydrates should be considered. There are also practical concerns such as the number of bacterial species used should not make the system too complex and the bacteria chosen should be compatible with each other. Furthermore, the media chosen for growth of the bacteria should be as simple as possible and relevant for the colon but at the same time assure good growth of all bacteria chosen. Below we describe five possible bacterial candidates and the rationale for choosing these candidates for a novel colon dissolution media, primarily for evaluating carbohydrate-based formulations.

*Faecalibacterium prausnitzii* ATCC 27768. The species *F. prausnitzii* belongs to the phylum *Firmicutes*, which, as mentioned before, is one of the phyla dominating the colonic microbiota. In a study performed by Hold et al. where fecal samples from 10 healthy individuals consuming a Western diet were collected, it was shown that *prausnitzii* made up between 1.4 and 5.9% of the total microbiota [154]. *F. prausnitzii* is one of the most important producers of the short chain fatty acid butyrate and therefore the species has an influence on human health [155]. In a study comparing fecal samples from 20 patients diagnosed with colorectal cancer, with 17 healthy individuals it was shown that the amounts of *F. prausnitzii* and *E. rectale* in patients with colorectal cancer were almost four times lower. According to a study performed by Lopez–Siles and colleagues, in an attempt to determine what substrate promotes the growth of *F. prausnitzii*, no growth was observed when using arabinogalactan and little or no growth on soluble starch. No fermentation was observed on Xylan, only a few of the used strains were able to grow well on inulin, and while most strains grew well on apple pectin, the same was not observed with citrus pectin. Additionally, most strains were able to grow on the host-derived substrate *N*-acetylglucosamine and d-glucosamine and d-glucuronic acid could be used by some strains [156]. This indicates that *F. prausnitzii* can switch between non-host substrates and host substrates which gives it a competitive advantage when dietary intake is reduced [61]. The fact that *F. prausnitzii* was able to compete for apple pectin as a substrate with two species found in human feces that have been reported to utilize pectin, namely *Bacteroides thetaiotaomicron* and *E. eligens* [157], suggests that *F. prausnitzii* might play a significant role in pectin fermentation in the colon of humans [156]. There are also studies available that indicate that *F. prausnitzii* can utilize prebiotics such as galactooligosaccharides for growth [158]. Additionally, in a study performed by Ramirez-Farias and colleagues, consumption of inulin-oligofructose increased the levels of *F. prausnitzii*. However, more research on the effect of inulin on growth of this species is necessary since no effect on counts of *F. prausnitzii* was observed as a result of higher inulin intake in yet another study [159].

*Ruminococcus bromii* ATCC 27255 *R. bromii* is an anaerobic, Gram-positive bacterial species that belongs to the *Firmicutes*. It has been reported to be a starch-degrading species. In a study performed by Ze et al. [160]. the ability of *R. bromii* to use different starches was compared to that of three other amylolytic species present in the human large intestine, namely *E. rectale*, *B*. *thetaiotaomicron* and *B. adolescentis*. It was noted that the starch utilization varied between the species depending on the type of starch and its pretreatment. Overall, *R. bromii* and *B. adolescentis* appeared to be more active than the other two species and they were notably more active in degrading raw or boiled resistant starches. Additionally, when all four species were grown together, it was observed that the starch utilization was higher compared to when the combinations with the other three species without *R. bromii* were co-cultivated. From the results, it was concluded that *R. bromii* was the most potent resistant starch degrader in the experiment. In the same study, the ability of the species to utilize starch metabolites was investigated. While not being able to utilize glucose, *R. bromii* could make use of breakdown products such as fructose and pullulan. It was also observed that some breakdown products created by *R. bromii* could be used by the three remaining species, indicating that there is a possibility of cross-feeding between the species used in this study [160]. There are also articles available that report an increase in *R. bromii* in individuals on diets with a higher amount of resistant starch. In a study from 2011, where fecal samples from obese men on different diets were compared, an increase in the number of *R. bromii* and *E. rectale* was observed when subjects were given a diet rich in resistant starch. It also appeared that a resistant starch-enriched diet stimulated an increase in species related to *R. bromii* and *E. rectale* [59]. A similar increase in *R. bromii*-related species was observed in a study using healthy individuals on diets supplemented with resistant starch. Additionally, despite the fact that an increase in SCFAs was noted in fecal samples when the subjects were on the resistant starch diet, this observation could not be connected to the abundance of *R. bromii* [161].

*Eubacterium rectale* ATCC 33656. As previously mentioned, *E. rectale* is a *Firmicutes* species belonging to the *Clostridia* cluster XIVa, that has been reported to be abundant in the human colon. Studies have shown that the species has an ability to degrade resistant starch and that its abundance in feces increases on a resistant starch-enriched diet [59]. Additionally, in a study from 2007, it was observed that the number of bacterial groups consisting of relatives of *E. rectale* and *Roseburia* spp. decreased significantly when subjects were on a diet low in carbohydrates. It was also noted that the concentration of SCFAs, and specifically butyrate, decreased in the feces in the same manner leading to the theory that these play an important role in butyrate production in the colon [162]. An *E. rectale* strain used in a study by Scott and colleagues, also proved to be able to grow on short chain fructooligosaccharides, xylooligosaccharides and amylopectin potato starch but failed to grow well on long chain inulin. The production of SCFA was also assessed in culture supernatants after 24 h of growth and showed butyrate, formate and lactate as the main fermentation products of *E. rectale* [163]. 

*Bacteroides uniformis* belongs to the *Bacteroidetes* phylum and allegedly has the ability to ferment a variety of polysaccharides. In a recent study from 2017, a strain of *B. uniformis* was isolated from infant stools, sequenced and grown on different carbon sources. Additionally, the genome expression patterns resulting from growth on these substrates were examined. The carbon sources used in the experiment were glucose, inulin, gum arabic (consisting of a complex mixture of branched polymers of galactose, rhamnose, arabinose, and glucuronic acid), pectin, wheat bran extract (mostly composed of a mixture of low weight xylo- and arabinoxylo-oligosaccharides) and mucin. It was observed that *B. uniformis* was able to grow on all these substrates. However, it was noted that the more complex polysaccharides such as those present in gum arabic, inulin and pectin were fermented less efficiently than the simpler oligosaccharides. It was concluded that this observation could be related to the origin of the bacterial strain, which in this case, came from infant stool. The analysis of the genome of *B. uniformis* showed that the species possesses a large panel of genes coding for so-called CAZymes (carbohydrate-active enzymes) compared to other *Bacteroides* species such as *B*. *thetaiotaomicron* and *Bacteroides cellulosyliticus*. Its ability to degrade mucin suggests that it might be able to grow on host-derived substrates and colonize the mucosa of the colon. Additionally, it was observed that production of the neurotransmitter GABA (gamma-amino butyric acid) increased significantly when *B. uniformis* was grown on mucin and pectin compared to growth on glucose. Lastly, pectin, gum arabic, and especially mucin, upregulated the expression of genes involved in butyrate production [164]. 

*Prevotella copri* CCUG 58058T. Even though *Prevotella* is mentioned as a family of bacteria that is common in the large intestine, it was difficult to find an example of a specific dominating species. Therefore, a random species known to exist in the large intestine was chosen, namely *Prevotella copri*. The species is a Gram-negative bacterium that belongs to the phylum *Bacteroidetes*. Scher and colleagues found that it is present in the stool of both healthy individuals and patients with rheumatoid arthritis, however, the species appeared to be overexpressed in the latter case. Since the prevalence of *P. copri* was similar in healthy individuals and in treated patients with reduced disease activity, they speculated that *P. copri* benefits from inflammatory conditions [165]. The fermentation of carbohydrates by *P. copri* in the large intestine does not seem to have been thoroughly explored. However, in a study from 2015, the gut microbiota was compared between healthy individuals who responded to the consumption of barley kernel-based bread, to those healthy individuals who did not respond or responded the least. Kovatcheva–Datchary and colleagues discovered an increased *Prevotella*/*Bacteroides* ratio in the responders compared to the non-responder. It was found that *P. copri* was the most abundant out of the *Prevotellaceae* species in the responders and this was also associated with an increased potential in fermentation of complex polysaccharides [166].

## 6. Conclusions

In this review, we have presented the most commonly used methods to study colon release. Although colon delivery has been studied extensively and there exists several different methods for in vitro studies of colon release and digestion of food, we think there is still a need for new in vitro methods. Primarily, there is a need for methods that include microorganisms but that are more reproducible and to some extent experimentally simpler than the current praxis of using fecal samples or caecum content.

## Figures and Tables

**Figure 1 pharmaceutics-11-00095-f001:**
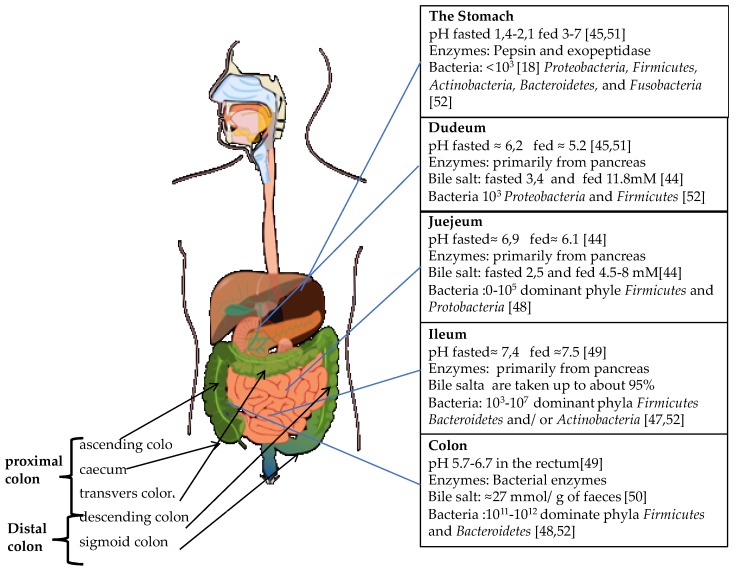
An overview of the conditions in the GI-tract with a focus on the composition of the liquid content. Data from References [18,50,51,53,54,55,56,57,58]. The picture of the GI-tract is by Mariana Ruiz, Jmarchn from Wikimedia commons. (https://commons.wikimedia.org/wiki/File:Digestive_system_without_labels.svg)

**Figure 2 pharmaceutics-11-00095-f002:**
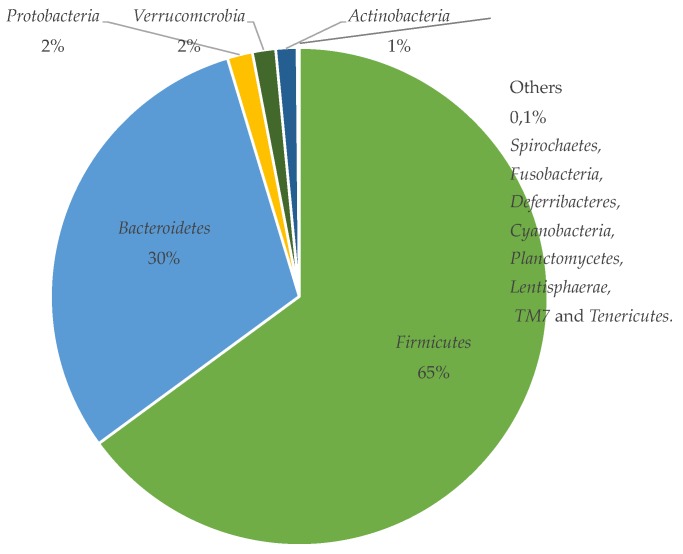
The phyla composition of the human gut from Yang et al. [148].

**Table 1 pharmaceutics-11-00095-t001:** Bile salt composition in the small intestines and colon. Data are from Ridlon et al. [52].

Bile Salt	Small Intestines	Colon
Cholic acid CA	35%	2%
Chenodeoxycholic acid CDCA	35%	2%
Deoxycholic acid (DCA)	25%	34%
Lithocholic acid LCA	1%	29%
Ursodeoxycholic acid UDCA	2%	2%
12-oxy-LCA		3%
Others	2%	28%

**Table 2 pharmaceutics-11-00095-t002:** Dissolution media in Pharmacopoeias and Guidelines resembling the upper GI tract.

GI Site Represented	Suggested Prandial State	Media	Complexity	References
Stomach	Fasted	pH 1.0–1.5	Low	[87,89,90,91]
		pH < 4.0 + Surfactant(s)	Medium	[79,87]
		pH < 4.0 + Enzymes (pepsin)	High	[87,89]
	Fed	Stomach pH + Physiological Surfactant(s)	Medium	[79]
No specific site	Fasted	pH 4.5	Low	[79,87,90,92]
	Fed	pH > 4.0 and < 6.8 + Enzymes (Papain)	High	[89]
Small intestines	Fasted	pH 5.5, 5.8, 6.5, 6.8 *, 7.2, 7.5	Low	[87,89,90]
		Intestinal pH + Physiological Surfactant(s)	Medium	[79]
	Fed	Intestinal pH + Physiological Surfactant(s)	Medium	[79]
		pH ≥ 6.8 + Enzymes (Pancreatic powder)	High	[87,89]

* pH 6.8 is the most frequently employed pH resembling the small intestines.

**Table 3 pharmaceutics-11-00095-t003:** Examples of colon bacterial enzymes used in simulated colon media for dissolution.

Type of Formulation Components	Enzyme	Refences
Azo-structures, polymers and conjugates	Azoreductase	[114,115]
Guar gum	Galactomannanase, α-galactosidase	[116,117]
Chitosane	β-glucosidase	[118,119]
Pectin	Pectinase	[120,121]
Starch	Amylase	[113]
Dextrane	Dextranase	[122]
Inuline	Inulase	[122]

**Table 4 pharmaceutics-11-00095-t004:** Species selected for the bacterial mixture, as well as the amounts of different bacteria used to create a dissolution media simulating the colon by Singh et al. [144] and Kotla et al. [146].

Singh, et.al., 2015 [140]		Kotla et al., 2016 [142]
Composition	Bacterial Count/Amount (CFU)	Composition
Lactobacillus acidophilus	0.75 × 10^12^	Lactobacillus acidophilus
Lactobacillus rhamnosus	0.75 × 10^12^	Lactobacillus rhamnosus
Bifidobacterium longum	0.75 × 10^12^	Bifidobacterium longum
Bifidobacterium bifidum	0.50 × 10^12^	Bifidobacterium infantis
Saccharomyces boulardii	0.10 × 10^12^	Lactobacillus plantarum
		Lactobacillus casei
		Bifidobacterium breve
		Streptococcus thermophilus
		Saccharomyces boulardii

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
