# Peer review of "In Vitro Methods to Study Colon Release: State of the Art and An Outlook on New Strategies for Better In-Vitro Biorelevant Release Media"

_pharmaceutics, 2019, doi:10.3390/pharmaceutics11020095_

Round 1

Reviewer 1 Report

This work is very well written, even if there are some concepts known for some time( e.i. the whole chapter from line 124 to line 155 does not tell anything new).Another advice that I would like to give to the authors is to improve the bibliography, as only 37 bibliographic entries out of 153 reference to works that go from 2015 to today. the others are much older. Please look for new concepts in literature and replace them with older ones.

Author Response

The work is very well written, even if there are some concepts known for some time (for example, the whole chapter from line 124 to line 155 does not tell anything new). Another advice that I would like to give to the authors is to improve the bibliography, as only 37 bibliographic entries out of 153 refererence to works that go from 2015 to today, the others are much older.. Please, look for new concepts in the literature and replace them with older ones. We appreciate the comments from the reviewers and have don our best to address these. 

We have shorten the text from line 124-155 and we have added newer references however we have only to a minor extent replaced the old ones as it is our experience that especially younger scientists are not fully aware of the earlier work done. 

Reviewer 2 Report

The review paper “In vitro methods to study colon release. State of the art and an outlook on new strategies for better in vitro biorelevant release media” from Marie Wahlgren et al., gives an overview of the most common methods applied to test enteric formulations. A particular focus is given to colon-targeted formulations, specifically, reliable methods for mimicking the colon environment are reviewed.

This work, in my opinion, is well-written and includes most of the literature on the topic. The treated arguments fall within the scope of Pharmaceutics and could involve a large audience. I suggest the acceptance of the present paper with minor revisions.

Authors’ addresses: please insert the full affiliation including address, city and country.

Line 49: as Inulin was included in the subsequent tables, it should be listed also here.

Line 56: since the term “wax” is included here for the first time, an explanation of the meaning should be given here. I mean, is wax intended to indicate natural waxes, petroleum based waxes or products like carbowax (PEG-based)?

Line 88-89: when mentioning the Pharmacopeia, the respective Country should be mentioned. In these two lines, which Pharmacopeia is referenced?

Line 112: please define the term “hydrodynamics” linking it to the sentence.

Line 251: the administration of probiotics has not been mentioned before in the text but I think it is an important aspect that is worth of a brief paragraph even at this point of the review.

Author Response

We appreciate the comments from the reviewers and have don our best to address these, see our comments below

Authors’ addresses: please insert the full affiliation including address, city and country. This has been included

Line 49: as Inulin was included in the subsequent tables, it should be listed also here. We have included inuline

Line 56: since the term “wax” is included here for the first time, an explanation of the meaning should be given here. I mean, is wax intended to indicate natural waxes, petroleum based waxes or products like carbowax (PEG-based)? We have clarified that this is about natural waxes

Line 88-89: when mentioning the Pharmacopeia, the respective Country should be mentioned. In these two lines, which Pharmacopeia is referenced? This has been clarified

Line 112: please define the term “hydrodynamics” linking it to the sentence. We have changed the text to make it more clear. 

Line 251: the administration of probiotics has not been mentioned before in the text but I think it is an important aspect that is worth of a brief paragraph even at this point of the review. We have added a a short paragraph on probiotics to the text

Reviewer 3 Report

Dear Authors,

 After the review process, I have several comments: 

- you should reformulate the following part from abstract ”... which forms a very complex ecosystem. The complexity of the microbial community ...” to avoid repetition; 

- you should explain why they included in the first section (pag. 2, line 45-53) only the polysaccharides, in vitro studies prove that there are many other active ingredients that are affected by the fermentative action of microbiota; 

- you should explain the possible biotransformations and influence of these processes in biological activity of the secondary metabolites; 

- you should make necessary correction in pag 4, line 166 - names of colon segments; the authors should correct ml with mL, in whole manuscript; 

- you should include references in pag. 7, line 300; 

- you should replace microflora with microbiota, in whole manuscript; 

- you should mention data which are based on the microbiota fingerprint modulation, media for fingerprint restoration, an essential phase for a relevant in vitro simulation; 

- you should include a conclusion paragraph which presents what is significant for readers and possible future research evolution in this domain.

Best regards!

Author Response

We appreciate the comments from the reviewer and have don our best to address these, see our comments below

- you should reformulate the following part from abstract ”... which forms a very complex ecosystem. The complexity of the microbial community ...” to avoid repetition; This has been changed.

- you should explain why they included in the first section (pag. 2, line 45-53) only the polysaccharides, in vitro studies prove that there are many other active ingredients that are affected by the fermentative action of microbiota; We have chosen to add two azo and carbohydrates as they are the most common formulation principles used for release through colon microbiota. We have clarified the text by adding the following two sentences One strategy is to use excipients that are depredated by the microorganisms in colon. Two of the most common classes of excipients used are carbohydrates and azo compounds

- you should explain the possible biotransformations and influence of these processes in biological activity of the secondary metabolites; We have chosen not to include this in the article as we find that this would be slightly out of scope for the current review. The review are more focused on in vitro release and not biological activity of the substances released.

- you should make necessary correction in pag 4, line 166 - names of colon segments; the authors should correct ml with mL, in whole manuscript;  Line 166 has been removed due to comments from another reviewer and ml has been changed to mL

- you should include references in pag. 7, line 300; References has been added 

- you should replace microflora with microbiota, in whole manuscript; This has been done

- you should mention data which are based on the microbiota fingerprint modulation, media for fingerprint restoration, an essential phase for a relevant in vitro simulation; We have added this to the paper

- you should include a conclusion paragraph which presents what is significant for readers and possible future research evolution in this domain. This has been added